# The Changing Role of Allogeneic Stem Cell Transplantation in Adult B-ALL in the Era of CAR T Cell Therapy

**DOI:** 10.3390/curroncol32030177

**Published:** 2025-03-19

**Authors:** Jana van den Berg, Claudia Meloni, Jörg Halter, Jakob R. Passweg, Andreas Holbro

**Affiliations:** 1Division of Hematology, University Hospital Basel, CH-4031 Basel, Switzerlandjakob.passweg@usb.ch (J.R.P.); andreas.holbro@usb.ch (A.H.); 2Innovation Focus Cell Therapies, University Hospital Basel, CH-4031 Basel, Switzerland; 3Regional Blood Transfusion Service, Swiss Red Cross, CH-4056 Basel, Switzerland

**Keywords:** allogeneic, hematopoietic transplantation, acute lymphoblastic leukemia, chimeric antigen receptor T cells, CD19, consolidation

## Abstract

The treatment of B-cell acute lymphoblastic leukemia (B-ALL) in adults remains a significant therapeutic challenge. While advances in chemotherapy and targeted and immunotherapies have improved overall survival, relapsed or refractory (r/r) adult ALL is associated with poor outcomes. CD19-directed chimeric antigen receptor (CAR) T-cell therapy has emerged as a transformative option, achieving high remission rates even in heavily pretreated patients. However, relapse is common. Allogeneic hematopoietic stem cell transplantation (allo-HCT), a traditional cornerstone of remission consolidation, may improve long-term outcomes but carries risks of transplant-related mortality (TRM) and morbidity. Most evidence for HCT after CAR T therapy comes from retrospective analyses of subgroups from CAR T cell trials, with small sample sizes and inconsistent data on transplant procedures and outcomes. Despite these limitations, consolidative allo-HCT appears to prolong relapse-free survival (RFS). While overall survival (OS) benefits are in question, extended remission duration has been observed. Nonrelapse mortality (including TRM), ranging from 2.4 to 35%, underscores the need for careful patient selection. Emerging real-world data affirm these findings but highlight the importance of individualized decisions based on disease and treatment history. This review examines current evidence on the sequential use of CD19-directed CAR T-cell therapy and allo-HCT in adults with r/r B-ALL.

## 1. Introduction

Treatment of adult B-cell acute lymphoblastic leukemia (ALL) remains a significant challenge. While advances in chemotherapy, and targeted and immunotherapies have improved overall survival, relapsed or refractory (r/r) adult ALL is associated with dismal outcomes. Prognosis is particularly poor in cases where minimal residual disease (MRD) persists after initial therapy and in patients who relapse [1]. Depending on disease type and treatment, up to 10% of patients are refractory to primary therapy, and up to 30% to 60% will relapse [2]. Furthermore, response rates decrease with each subsequent line of salvage therapy from up to 45% after the first salvage to 14% in the third or greater salvage treatment, underscoring the need for innovative strategies to achieve deeper remissions [3]. Although novel agents, such as blinatumomab, have improved this, the response is usually not durable unless allogeneic hematopoietic stem cell transplantation (allo-HCT) is performed [4,5]. Historically, allo-HCT has been a cornerstone in consolidating remission in high-risk ALL patients. Its curative potential lies in the graft-versus-leukemia effect, but it is also associated with considerable risks, including graft-versus-host disease (GvHD) and transplant-related mortality (TRM). In recent years, CD19-directed chimeric antigen receptor T-cell (CAR T-cell) therapy has revolutionized the treatment of r/r B-cell ALL, especially for pediatric patients. By harnessing the cytotoxic potential of genetically modified autologous T-cells, CAR T therapy induces high rates of complete remission (CR), of up to 90%, even in heavily pretreated patients (Table 1) [6,7,8]. Despite these successes, the durability of response remains a concern, with relapse often driven by antigen escape or limited CAR T-cell persistence. The combination of CAR T therapy and consolidative allo-HCT presents an intriguing strategy for optimizing long-term outcomes. While CAR T therapy alone has the potential to achieve long-term remissions, the role of subsequent allo-HCT remains unclear. The potential benefit of allo-HCT must be weighed against the substantial risks of transplant-related morbidity and mortality (TRM), raising the key question: Who benefits most from allo-HCT after CAR T therapy, and can certain subsets of patients achieve durable remission without transplantation? Most of the evidence regarding allo-HCT following CD19-CAR T cell therapy is derived from retrospective analyses of clinical trial outcomes. Focusing on the efficacy of CAR Ts, however, transplant indications, conditioning regimens, and transplant-specific outcomes are often not comprehensively reported. Regarding its indications, allo-HCT is often used as consolidation but may also serve as a salvage strategy for relapse. In this review, we examine the current evidence on the role and outcomes of allo-HCT as consolidation after CD19-directed CAR T-cell therapy in adults with r/r B-cell ALL.

## 2. Materials and Methods

A structured search of PubMed and Web of Science was conducted to identify studies (published in English up to 1 December 2024) evaluating allo-HCT following CD19-directed CAR T-cell therapy in adult B-ALL. Keywords included ‘B-ALL’, ‘relapsed/refractory ALL’, ‘CAR T-cell therapy’, and ‘allogeneic hematopoietic stem cell transplantation (allo-HCT/HSCT)’. Retrospective analyses, registry-based studies, and clinical trial data were reviewed. Only anti-CD19 CAR T studies were included, while products targeting other epitopes (e.g., CD22) were excluded. FDA-approved CAR T products were analyzed separately, irrespective of the search results. While pediatric-only studies were excluded, studies on mixed cohorts were evaluated, with age ranges specified for each study. This review is based on publicly available literature. Reference lists of relevant articles were screened for additional sources.

## 3. Main

### 3.1. Allo-HCT as Consolidation After CD19-Directed CAR T-Cell Therapy—Review of the Literature

Evidence regarding HCT following CD19-CAR T-cell therapy is mostly derived from small retrospective analyses of clinical trial outcomes. Focusing on the efficacy of CAR Ts, however, the patient populations, the rationale to proceed to transplant, details on the transplant procedure (e.g., conditioning, GvHD [prophylaxis]) and HCT-specific outcomes are often heterogeneous and inconsistently detailed in the publications. Historically, in early trials involving adults, allo-HCT was considered the standard of care, and patients achieving CR after CAR T were often referred for transplantation [6,9,24,25,26,27]. Studies by Brentjens, Davila, and colleagues highlighted this practice, stating that “for ethical reasons, and as per the protocol, after adoptive T cell therapy, eligible patients underwent subsequent allo-HCT” [25,26].

In the preliminary report from 2013 on NCT01044069, in which adult patients with relapsed B-ALL received CAR T cells targeting CD19 with a CD28 costimulatory domain, Brentjens et al. reported on the first 5 patients. All achieved MRD-negative remission, and four proceeded to a first consolidative allo-HCT. The one patient who was ineligible for transplantation because of pre-existing comorbidities relapsed 90 days after CAR T infusion [25]. In an updated report (2014; n = 16 patients) by Davila et al., 4 of the 16 patients had previous allo-HCT. 14 (88%) reached morphologic CR or CRi, with MRD-negativity in 12 patients. Seven out of 10 were considered eligible (CR without contraindicating comorbidities) to proceed to transplantation. All had persistent remission (post-allo-HCT follow-up: 2–24 months), but two patients died from transplant-associated complications [26]. Finally, having treated 53 adult patients who were heavily pretreated, including 19 (36%) with prior allo-HCT and 19 (36%) with primary refractory disease, the group published long-term follow-up (median of 29 months). Furthermore, 44 patients (83%) achieved CR after 19–28 z CAR T cells (with MRD-negativity in 32 patients). Of those, 17 (39%) went on to allo-HCT, after which 6 (35%) died from transplant-related toxicities, 6 (35%) relapsed, and 5 (30%) continued to be in complete remission. Of the 26 (59%) patients not proceeding to allo-HCT, 17 had a relapse or died and 9 (35%) were alive during follow-up. Hence, Park et al. found no significant difference in event-free and overall survival between patients who underwent allo-HCT and those who did not. The group highlights that for long-term response, pre-CAR T disease burden was the best predictor of remission duration and survival, with a median overall survival of 12.9 months globally, but 20.1 months (95% CI, 8.7 to not reached) for patients with a low disease burden (defined as <5% bone marrow blasts) before treatment. This significant difference was found to be independent of allo-HCT after CAR T-cell infusion [6].

Pan et al. treated a mixed cohort of 51 (younger) adults and pediatric patients (median <18 years, range 2–68) with CD19-directed CARs (low-dose; containing a 4-1BB co-stimulatory domain) with the aim to induce remission and proceed to transplantation. Prior to CAR infusion, 9 patients were MRD-positive on flow-cytometry (FCM-MRD+; median age 24) and 42 were in morphologic relapse or primary refractory (r/r; 6.5–98.5% blasts in the bone marrow; median age 11). Two patients died within the first three weeks of infusion (from progression with coagulopathy and infection). Of the 45 patients who responded to CAR T-cell therapy, 34 of the evaluable 40 patients (85%) in the r/r cohort and all 9 in the FCM-MRD+ group achieved FCM-MRD-negative status. A total of 27 out of 45 responders (60%) proceeded to allo-HCT. Among these, two patients died from TRM, and three relapsed. Of the 18 patients who decided not to receive allo-HCT, 9 relapsed after a median time of 64 days (range 40–245 days). The 6-month relapse rate after HCT was calculated to be 11.9% with a 6-month TRM of 7.5%. Due to the significant difference in relapse rate, Pan et al. suggested a potential benefit of consolidative allo-HCT following CAR T-cell therapy [9]. This finding was repeated in a similar trial with small patient numbers (34 patients enrolled, 11 received allo-HCT) on CD22-directed CARs for patients who failed previous CD19 CAR T-cell therapy [24].

Likewise, referring to the standard of care for patients to proceed to transplantation if they achieve remission, in the study by Hay et al. on treating 53 patients with CD19-CARTs containing a 4-1BB costimulatory domain, 18 (40%) of 45 MRD-negative patients underwent HCT. While overall, 22 (49%) of those 45 patients relapsed (at a median of 3.5 months after CAR T-cell infusion), the cumulative incidence of relapse within the transplanted cohort was much lower (17%), with an overall two-year EFS of 61%, OS of 72% (median follow-up 28.4 months after allo-HCT) and a nonrelapse mortality of 23%. Moreover, 19 of 27 responders to CAR T who did not undergo transplantation relapsed. Acknowledging differences between patients who did or did not receive transplants, such as a lower disease burden before CAR T therapy or fewer previous allo-HCTs in the transplant cohort, the group concludes that allo-HCT may still improve EFS. The decision to perform transplantation or not was not stated per individual patient, but authors refer to the standard of care with parameters such as comorbidities, age, history of prior transplantation and patient preference [10].

In a similar trial, Jiang et al. treated 58 patients with anti-CD19 CAR-T cells, of whom 47 achieved MRD-negative CR by FCM. Out of these, a cohort of 21 received consolidative allo-HCT at a median of 44 days after CAR T, whereas 26 patients did not. Reasons to defer transplant were previous allo-HCT (n = 3), contraindications (i.e., comorbidities/infections; n = 5), the lack of a suitable donor (n = 3), and personal reasons (willingness, economic factors and life quality; n = 15). Overall, for all CAR T treated patients, 22/58 (38%) patients relapsed (at a median of 3.3 months after infusion). Of the 21 patients transplanted, two relapsed (10%; at days 57 and 261 after allo-HCT) and two died in CR of transplant-related complications (GvHD and infection). Including a landmark analysis, authors found that relapse-free survival (RFS) was significantly prolonged by allo-HCT, especially in higher disease burden pre-CAR T infusion, but no difference in overall survival [11].

In the pivotal trials leading to the approval of brexucabtagene autoleucel (brexu-cel, formerly KTE-X19), an anti-CD19 CAR T-cell therapy with a CD28 costimulatory domain, allo-HCT was permitted as a consolidative therapy at the physician’s discretion but was not protocol-defined [12]. In a post hoc analysis including 78 patients who received the pivotal dose of brexu-cel in phases I (n = 23) and II (n = 55) of the ZUMA-3 trial, Shah et al. sought to explore the impact of prior therapies and subsequent transplantation. Additionally, 14/57 (25%) of patients who reached CR or CRi proceeded to allo-HCT, after a median of 95 days (range 60–390) and 134 days after infusion, respectively. At the time of analysis (median follow-up of 29.7 months), 7 of the 14 (50%) patients who received allo-HCT were in ongoing remission, 1 (7%) patient had relapsed, and 5 (36%) patients had died. Of the 43 responders without subsequent transplantation, 12 (28%) remained in remission, 19 (44%) relapsed and 3 (7%) died. The median duration of remission (DOR) was 44.2 months in those receiving allo-HCT compared to 18.6 months in responders without transplantation, but overall survival (OS) was similar: 47.0 months with allo-HCT vs. not reached. As suggested in phase II results, brexu-cel led to favorable outcomes in adult patients with r/r B-ALL and may induce durable responses alone in some patients [12,28]. Accordingly, and based on overall survival data, the authors conclude that subsequent allo-HCT was not necessary to achieve a survival benefit. However, patients with subsequent allo-HCT appeared to have a longer response duration [12].

Tisagenlecleucel (tisa-cel, formerly CTL019), an anti-CD19 CAR T-cell therapy with a 4-1BB costimulatory domain, was more extensively tested and later approved for ALL in children and young adults (ELIANA trial [29]). However, in the first in-human trial (NCT01029366 [7]), 5 adults (median age 47) were included next to the 30 young patients, with a follow-up trial (NCT02030847) that enrolled 42 adults, of which 30 received tisa-cel (through 3 dose cohorts) [13]. At a median follow-up of 13 months, Frey et al. reported on the combined outcomes of these 35 adults. For all of those who achieved CR, defined as <5% bone marrow blasts (24/35, 69%), the median OS was not reached, with a 2-year survival rate of 64% and a median EFS of 19.4 months. Nine of 24 patients who achieved CR after CAR Ts proceeded to consolidative allogeneic HCT after a median time of 2.6 months (range 1.7–5.2), the other 15 received no other treatment unless relapse occurred. In a landmark analysis, a significant improvement in EFS and a nonsignificant improvement in OS were found for patients who underwent HCT. Despite this survival trend, the authors concluded that durable remissions were also seen after CAR T in patients achieving a CR [13].

For the CAR T product most recently FDA-approved, obecabtagene autoleucel (obe-cel, formerly AUTO1, a 41BB-ζ anti-CD19 CAR that utilizes a distinct single-chain variable fragment), the rate of patients to proceed to consolidative HCT was lower than in other trials [22,23] (Table 1). As reported by Roddie et al., 3 out of 17 (18%) in Phase I and 18 out of 99 (18%) in Phase II received HCT after responding to obe-cel. Overall, in the phase 2 FELIX trial, 77% of patients responded (CR/CRi) to CAR T, and 12-month EFS and OS estimates were 49.5% and 61.1%, respectively. With transplant-related outcomes not stated in detail, no substantial difference in EFS or OS was reported for patients who did (n = 18) or did not receive allo-HCT (median follow-up 21.5 months) [23].

In a study by Aldoss et al. (2023) on memory-enriched CD19-directed CAR Ts (with CD28 costimulation), 40 out of 42 evaluable adult patients responded (CR/CRi rate of 95%; FCM-MRD–negative 95%) to CAR Ts, of which 21 responders proceeded to allo-HCT while in remission. For the 21 patients who underwent consolidative transplant, 2 patients relapsed (10%) and 1 died of infection 2.5 months after allo-HCT, whereas of the 19 responders without allo-HCT 11 (58%) relapsed within a median time of 3.4 months post CAR T. The authors concluded that there is a significant benefit in RFS in responders who underwent allo-HCT consolidation at a low nonrelapse mortality rate (1 death (5%) within 100 days). In an exploratory univariate analysis, among others, age, intensity of bridging therapy, prior exposure to blinatumomab and inotuzumab, and failing prior allo-HCT were not associated with RFS [14]. In a separate retrospective study, Aldoss et al. (2024) analyzed outcomes of 45 patients who responded to CD19-targeted CAR-T therapy (44% commercial (24% brexu-cel, 20% tisa-cel) and 56% investigational products) and received allo-HCT. With a median long-term follow-up of 2.47 years (range: 0.13–6.93), survival outcomes of allo-HCT consolidation were found to be favorable with relatively low early transplant-related mortality. The cohort was heavily pretreated, with a median number of 5 prior lines and 35% of patients in CR3 or beyond, yet 2-year OS, RFS, and the cumulative incidence of relapse (CIR) were 57.3%, 56.2% and 23.3%, with no significant difference between patients who underwent their first vs. second transplant. Overall NRM at day 100 and 2 years was 2.4% and 20.4%, and 2 years NRM was not statistically different for first vs. second transplant (12.5% vs. 32%, *p* = 0.120), respectively. With an objective to compare patients with first to second transplant, no significant difference in 2-year OS (52% vs. 68%, *p* = 0.641) or RFS (54% vs. 59%, *p* = 0.820) was found, despite the use of RIC and NMA conditioning as opposed to MAC regimens in 95% of patients receiving their second allo-HCT [15].

While retrospective analyses of CAR T trials included relatively small subsets of patients who received transplantation, real-world and registry studies, despite being retrospective, offer the advantage of larger cohorts. Wudhikarn et al. analyzed outcomes from the CIBMTR registry (Center for International Blood & Marrow Transplant Research). The investigators evaluated the effectiveness of brexu-cel in this cohort of 242 patients, finding overall results consistent with those observed in ZUMA-3 in a broader patient population. While specific results following consolidative allo-HCT were not (yet) reported, the practice remained similar to the pivotal trial, with 29% of responders undergoing allo-HCT after CAR T cell infusion at a median of 107.5 days post-treatment [16].

Roloff and colleagues of the ROCCA consortium (31 CAR T-cell centers in the US) recently reported on more detailed outcomes of 189 patients who received brexu-cel outside the context of clinical trials by retrospective data analysis (median follow-up of 11.4 months). With a median of four lines of prior therapy, patients were heavily pretreated, and 78 (41%) had undergone previous HCT. At apheresis, 79 (42%) were in morphologic remission, of whom 27% were MRD+/unknown and 15% were MRD-negative. This was different from the ZUMA-3 population, where participants were required to have >5% blasts at screening. Overall, of 168 patients evaluable at day +28 after CAR T infusion, 151 (90%) achieved CR/CRi, with MRD-negative CR in 119/151 (79%). Of 151 patients who achieved CR after brexu-cel, 47 (31%) relapsed. With a median follow-up of 11.4 months, the PFS rate at 12 months was 48% (46% after censoring for consolidative allo-HCT), with an unadjusted median PFS of 9.5 months. OS rate was 63% (identical after censoring for allo-HCT), and median OS was not reached. Thirty patients, of whom only four had prior transplantation, received allogeneic HCT as consolidation at a median time of 99 days after brexu-cel (range, 45–234). In a multivariable analysis for the association of treatment-specific characteristics with PFS, these authors found that postremission consolidative transplantation was associated with superior PFS (HR, 0.34 [95% CI, 0.14 to 0.85]). Among the 26 patients who underwent allo-HCT in MRD-negative CR, 12-month PFS and OS rates were 74% and 77%, respectively, i.e., better than the full CR/CRi cohort censored for HCT: 46% and 63%, respectively. A total of 13% died of NRM overall, and 17% of transplant recipients (5/30) died of HCT-related complications [8].

Very few studies to date, all nonrandomized, explicitly investigate the effect of allo-HCT on outcomes after achieving CR through CAR-T cell therapy. The trial by Jiang et al., described above, had the objective to explore the effect of consolidative allo-HCT, but in their nonrandomized pragmatic trial (PCT), the allocation to the transplant vs. nontransplant cohorts followed disease status as well as personal reasons in a pre-HCT evaluation after CAR T therapy [11]. In contrast, while factors for cohort allocation were similar (patients’ willingness, donor availability and economic factors), Han et al. prospectively divided the patients into two groups before CAR T infusion. The median follow-up time was 398.5 days (range, 105–1530). However, as with most available studies, the cohort size remained small. Thirteen of fourteen patients in the prespecified HCT group achieved remission and proceeded to transplant. Additionally, 17 of 23 patients in the non-HCT group were responsive to CAR T and continued with follow-up. Of this, 6 patients out of 13 (46%) experienced relapse after transplantation, compared to 12 out of 17 (71%) patients in the non-HCT group, resulting in cumulative relapse rates of 46.15% and 70.59%, respectively. In the HCT group, two patients died from TRM (1 aGvHD, 1 transplant-associated thrombotic microangiography (TA-TMA)) and one patient died of severe infection. Tumor-related mortality was 38.46% (5/13) in the HCT group and 64.71% (11/17) in the non-HCT group. The median LFS was 557 days in the HCT group versus 238 days in the non-HCT group, and the median OS was 930 days in the HCT group compared to 384 days in the non-HCT group. Despite a tendency towards superior OS and LFS in the HCT group, no statistical significance was reached [17].

In a pediatric and young adult cohort (median <18 years, age 1–25), Summers et al. specifically sought to retrospectively investigate the impact of consolidative allo-HCT in a subset of patients from their phase I and II trials on CD19-directed CARs with a 4-1BB costimulatory domain. Fifty subjects in an MRD-negative remission beyond day 63 after CAR infusion were included in the analysis, excluding five who did not respond, eight who relapsed, and one who died before day 63. Twenty-three patients (median age 12 years), of whom 10 (43%) had received previous HCT, proceeded to HCT at a median of 3 months after CAR T infusion (range 2–9.9 months). Five of the twenty-three patients (22%) who underwent transplantation experienced relapse, and one died from TRM (pulmonary hemorrhage). On the other hand, 19 of the 27 subjects (70%) who did not undergo HCT had leukemic relapse. Of note, the non-HCT group included more patients who had prior HCT before CAR T (24/27(89%) with prior HCT vs. 10/23(43%) without). Overall, hence, LFS was found to be superior in subjects who underwent HCT (HR, 0.31; *p* = 0.01). LFS also tended to be superior in a subgroup analysis of patients with no prior HCT who were now transplanted or followed up despite a small sample size: 1/13 patients relapsed after the first transplant, as opposed to 2/3 in follow-up. From comparing only subjects who had a prior history of HCT, i.e., for whom this was the second transplant or follow-up only (10 vs. 24), no significant difference was found. Hence, authors conclude that the overall benefit of consolidating HCT was derived from those without a history of HCT, but which may have confounding factors, such as intensity of conditioning (more intense and likely to include TBI in the first allo HCT) [30].

### 3.2. Risk Factors for Relapse After CAR T Therapy

To date, several risk factors for relapse after CAR T therapy have been identified and include pre- and postinfusion factors, although the first may be more helpful for timely referral and preparation for transplant. Preinfusion risk factors include a high disease burden defined as no morphologic CR (≥5% bone marrow blasts) or extramedullary disease (EMD) [6,16,31,32]. The latter was recently examined in a large dataset of young patients (105 out of 300 who had relapsed after tisa-cel before HCT), indicating that 1 in 4 post-CAR T relapses involve EMD sites, of which two-thirds had a history of EMD before CAR, and outcomes are very poor with a median survival of less than 1 year [32]. Further risk factors include high-risk disease (such as KMT2A rearrangement and Ph-like ALL [2,33,34], the number of previous therapies and previous immunotherapy with blinatumomab [12,35] and/or inotuzumab-ozogamicin [8]. With contrasting results, the sole need for bridging therapy may not be an independent predictor as it is likely confounded by selection bias favoring high-risk disease and by heterogeneity in the type and intensity of bridging strategies. While the larger ROCCA dataset found bridging to be associated with inferior survival [8], others found an improved EFS by reduction in leukemia burden with bridging therapy [36]. Postinfusion risk factors includes the detection of next-generation sequencing (NGS)-MRD, which has been found to be highly sensitive for identifying patients at risk of relapse when detected in bone marrow following CD19-CAR T, but is not yet routinely used and standardization of MRD assessment is needed [8,37]. Another is an early loss of B cell aplasia (BCA, typically defined as occurring within 6 months of infusion). BCA has been shown to correlate with reduced CAR T-cell persistence and higher relapse rates [7,29], with a significant LFS-benefit for patients proceeding to allo-HCT as opposed to patients who did not [30].

## 4. Discussion and Conclusions

Despite significant progress, the treatment of adults with r/r B-ALL remains challenging. CD19-directed CAR T-cell therapy has revolutionized the therapeutic landscape, offering unprecedented remission rates of up to 90% [6,7,8,28]. However, achieving durable remissions remains an unmet need, with more than half of the patients experiencing relapse in the long term [31]. Historically and within current guidelines, allo-HCT is considered an effective treatment for preventing relapse. In clinical practice, most centers assess the indication for allo-HCT based on MRD status/response, and highlight that allo-HCT should be considered in all patients with r/r ALL [2]. Unfortunately, transplant-related mortality remains of significant concern, with rates reported as high as 13% following allo-HCT from HLA-matched sibling donors and 21% for transplants from unrelated donors [2,38]. Today, with emerging evidence that long-term remission may be achieved in a subset of patients with CAR T alone, the question of consolidation after CAR T must be weighed against the risks of TRM and poses a major challenge. Further key considerations for optimizing outcomes include identifying who benefits most from allo-HCT, how to assess relapse risk—where some evidence is available—and the influence of transplant-related factors such as patient and donor selection, as well as conditioning regimens, for which very limited data exist. While outcomes with HCT in pediatric patients seem favorable [39,40], results for adults varied, with a lack of randomized comparisons. However, reviewing the studies described above (and Table 1), more seemed to find a benefit in consolidating remission with allo-HCT. However, caution on interpretation is warranted since most studies were not designed to answer this question. Landmark analyses, to minimize a possible bias by allo-HCT being a delayed intervention [41], were rarely performed, namely (with limitations) in 4 out of the 18 studies reviewed in Table 1 [9,11,13,14]. Furthermore, sample sizes are small and the populations are heterogeneous. While on average 38% (range 11–78%) of responders to CAR T therapy received allo-HCT, the average absolute number of patients per trial is 18 (range 3–56) (Table 1). Information on conditioning regimens for HCT is only available for a few studies in which more than 3/4 of patients have received myeloablative conditioning (MAC) regimens [9,10,11,17]. MRD status at the timepoint of transplantation is also mostly unknown. While possibly the equivalent, a response to CAR T is reported from assessment at approximately 1 month after infusion, while transplantation was performed on average approximately 2.5–3 months, with a range extending up to a year post-CAR infusion (Table 1). This is relevant, since MRD-status correlates with outcome after HCT [2,42]. Additionally, in a small cohort of 19 adult ALL patients, a longer interval between CAR T infusion and allo-HCT (>80 days compared to <80 days) was associated with an increased risk of death overall and a higher NRM, but MRD status for individual patients was not reported. Overall, however, adverse events for allo-HCT after CAR T were evaluated as not disproportionately enhanced as compared to known post-allo-HCT benchmarks [21].

In their meta-analysis, also clearly limited by small sample sizes and nonrandomized study designs, Hu et al. included pediatric and adult patients in CR after CAR-T cell therapy with or without subsequent HCT. Subgroup analysis of similar reports suggested a benefit of allo-HCT after CAR T, demonstrating lower relapse rates across all age groups [40]. While some describe a trend towards superior OS and LFS after HCT [17], Park et al. reported high relapse and TRM rates following HCT (6 out of 17 patients relapsed (35%) and 6 died from TRM (35%)), but they clearly stated no significant difference in event-free and overall survival [6]. No difference was also reported by Roddie et al., but with transplant-related outcomes not stated in detail [23]. On the other hand, Hay and colleagues utilized a multivariable modeling approach and found that patients undergoing transplantation after CAR T-cell therapy did have longer EFS [10], as did Frey and colleagues in their landmark analysis [13]. As far as evaluable, however, most others seem to describe at least a longer duration of response or superior PFS with postremission consolidative HCT [8,9,11,12,14], including recent real-world evidence [8]. Thus far, however, while some saw a trend or nonsignificant improvement, none of the available studies found a clear survival benefit. It remains unclear whether this is due to the small sample size, confounding factors (such as patient selection or prior HCT), a short follow-up, or the associated TRM.

Focusing on the effects of CAR Ts, details on TRM regarding HCT, and other important transplant factors such as conditioning regimen and donor constellations are often missing or incomplete. Of the studies reviewed (and Table 1), eleven reported NRM, with rates between 2.4% and 35%. This finding is similar to a general patient population undergoing allo-HCT in ALL (13–30% [2,21]) and reassuring, considering the relapsed/refractory and therefore highly pretreated situation. Judging from the limited cases described, there appeared to be no increased rates of GvHD, though further evidence is necessary to confirm this observation [21].

In pivotal studies and with available data, very few patients have been reported to undergo a second allo-HCT following CAR T-cell therapy for relapse after their initial allo-HCT. For example, in the combined analysis of phase I and II of ZUMA-3, 29 of the 78 patients that were treated had relapsed after allo-HCT, but only one received a second transplantation after CAR T [12]. Others, like Jiang et al., excluded patients with previous HCT from transplantation, whereas in NCT02146924, 10 out of 21 patients received HCT for the second time [14]. Interestingly, Roloff and colleagues found a strong correlation between previous HCT and superior PFS after brexu-cel in their real-world cohort [8]. Together with findings from Aldoss et al. (2024) who found no significant differences in OS, RFS or NRM between patients who underwent their first vs. second transplant, one might hypothesize that also patients with relapse after allo-HCT might benefit from consolidative allo-HCT after CAR T and that both reduced-intensity conditioning (RIC) and MAC regimens can be used [15].

To date, several risk factors for relapse after CAR T therapy have been identified, encompassing both pre- and postinfusion factors described above, including high disease burden, high-risk genetic subtypes and postinfusion NGS-MRD positivity, and early loss of BCA. However, evidence remains limited, and there is a critical need to refine risk-adapted strategies, particularly regarding how to act on these predictors. For example, while early BCA loss has been associated with increased relapse risk, its role in guiding transplant decisions remains unclear. Furthermore, NGS-MRD is not yet routinely used, and standardization of MRD assessment is needed to effectively integrate it into post-CAR T decision-making.

In conclusion, the integration of CAR T-cell therapy and allo-HCT offers a promising therapeutic pathway for adult patients with r/r B-ALL. While CAR T-cell therapy alone may suffice for some, consolidative allo-HCT remains beneficial for high-risk subsets, particularly those with persistent MRD. However, identifying which patients benefit most from allo-HCT remains a key challenge, with factors such as disease burden, MRD status, prior therapies, and CAR T-cell persistence playing a potential role in risk stratification. Although several studies suggest a benefit of consolidative allo-HCT after CAR T, interpretation is limited by small sample sizes, retrospective designs, and inconsistent reporting of transplant procedures and outcomes. Nevertheless, as for an earlier trial with a separate CAR T product [21], an expert consortium that analyzed real-world data on the use of brexu-cel, recommends HCT for eligible, in particular transplant naïve patients who achieve remission after CAR T [8]. Further studies are urgently needed to clarify the role and timing of allo-HCT post-CAR T therapy, ideally through prospective randomized trials. However, recognizing the challenges posed by the large sample sizes required for meaningful conclusions, larger retrospective (and) registry studies might offer a more feasible alternative. These studies could also leverage the advantages of broader cohorts to mitigate the impact of heterogeneity in patient populations and clinical practices. For the time being, an individualized approach considering disease factors, prior treatments and response (including MRD), as well as age, comorbidities, and donor availability, will guide the decision for consolidative allo-HCT after CAR T-cell therapy in adult patients with B-ALL.

## Figures and Tables

**Table 1 curroncol-32-00177-t001:** Overview of studies reporting on consolidative allogeneic stem cell transplantation after CAR T in adult r/r B-ALL.

Study	CAR Construct	CAR Conditioning	Patients Treated (n)	Age, Median (Range)	Previous Therapies, Median (Range), Prior Allo-HCT, n (%)	OS Full Trial, % (mo) or Median OS	CR Rate Full Trial % (n)	Patients w. Subsequent HCT, n (% of Responders)	HCT Conditioning (%)	Median Time to HCT, Days (Range)	OS Post-HCT, % (mo) or Median OS	Relapse * Without HCT, % of Responders (n)	Relapse After HCT, % of Responders (n)	TRM or NRM % (n)
Park et al. [6]	CD19.CD28	Cy or Flu/Cy	53	44 (23–74)	3 (2 to ≥4), prior allo: 19 (36%)	median OS 12.9 mo	83% (44/53)	17/44 (39%)	NR	74 (44–312)	NR	65% (17 **/26)	35% (6/17)	TRM: 35% (6/17)
Pan et al. [9]	CD19 41BB	Flu/Cy	51	11 and 24 *** (2–68)	NR	NR	90% (36/40) and *** 100% (9/9)	27/45 (60%)	MAC (100%)	84 (35–293)	NR	50% (9/18)	11.9% (3/25) ****	TRM: 7.5% (2/27) ****
Hay et al. [10]	CD19.41BB	Flu/Cy or Flu/Eto	53	39 (20–76)	3 (1–11), prior allo: 23 (43%)	median OS (for responders) 20 mo	85% (45/53)	18/45 (40%)	MAC (67%) RIC (33%)	70 (44–138)	72 (24)	70% (19/27)	17% (3/18)	NRM: 23% (4/18)
Jiang et al. [11]	CD19.41BB	Flu/Cy	58	28 (10–85)	3 (2–4), prior allo: 3 (5%)	61.1% (12) median OS 16.1mo	81% (47/58)	21/47 (45%)	MAC (100%)	44 (33–89)	NR	61.5% (16/26)	10% (2/21)	TRM: 10% (2/21)
Shah et al. [12]	CD19.CD28	Flu/Cy	78	42.5 (18–84)	2 (1–8), prior allo: 29 (37%)	median OS 25.4 mo	73% (57/78)	14/57 (25%)	NR	95 (60–390) *****	median OS 47mo	19/43 (44%) + 3 (7%) died	1/14 (7%) + 5 (36%) died	NR
Maude et al. [7]	CD19.41BB	43% Flu/Cy Others	30 (25 CAYA/5 adult)	14 (5–60)	2 (range NR), prior allo: CAYA 18 (72%) adults: 0 (0%)	78% (at 6 mo)	90% (27/30)	3/27 (11%)	NR	NR	NR	7/24 (29%)	0% (0/3)	TRM: 0% (0/3)
Frey et al. [13]	CD19.41BB	71% Cy 14% Flu/Cy	35	33.8 (21–70)	1–2: 12 (34%) 3–4: 13 (37%) ≥5: 10 (29%) prior allo: 13 (37%)	median OS 19.1 mos	69% 24/35	9/24 (36%)	NR	79 (51–158)	NR	NR	NR	NR
Aldoss et al., 2023 [14]	CD19.CD28	Flu/Cy	46	38 (22–72)	4 (1–9), prior allo: 29 (63%)	63.2% (12) (responders only)	40/46 (87%)	21/40 (53%)	NR	79 (50–192)	NR	58% (11/19)	10% (2/21)	TRM: 5% (1/21)
Aldoss et al., 2024 [15]°	CD19.*various*	44% Flu/Cy and NR	45	31 (19–67)	5 (2–7) prior allo: 19 (42%)	57% (24) (all had HCT)°	NA°	45/45 (100%)°	MAC (overall 49%; 1st allo 81%)° RIC/NMA (overall 51%; 1st allo 19%	93 (42–262)	57% (24)° 1st allo: 52% 2nd allo: 68%	NA°	23.3% 1st allo:33.5% 2nd allo: 8.5%	NRM d100/2years 2.4%/20.4%
Wudhikarn et al. [16]	CD19.CD28	NR	242	46.8 (18–84)	NR, prior allo: 32%	80% (6)	80%	29%	NR	108 (84–157)	NR	NR	NR	NR
Roloff, et al. [8]	CD19.CD28	93% Flu/Cy 7%other	189	46 (18–81)	4 (2–12), prior allo: 78 (41%)	63% (12)	90% (151/168)	30/151 (20%)	NR	99 (45–234)	74% (12)	NR	NR	NRM: 17% (5/30)
Han, et al. [17]	CD19 4th-gen (CD28/CD27/CD3ζ-iCasp9)	Flu/Cy	37	32.5 (18–68)	≥4 in >85% prior allo: 0%	median OS (without allo group) 12.6mo	81% (30/37)	13/30 (43%)	MAC (100%)	64 (34–153)	median OS (with allo group) 30.5 mo	71% (12/17)	46% (6/13)	TRM: 15% (2/13)
Wang et al. [18]	CD19.41BB	Flu/AraC	23	42 (10–67)	2 (2–3) prior allo: 0%	median OS (without allo group) 10mo	87% (20/23)	5/20 (25%)	NR	NR (28–84)	median OS (with allo) not reached (FU 14mo)	7/18 (39%)	0% (0/5)	TRM: 20% (1/5)
Jacoby et al. [19]	CD19.CD28	Flu/Cy	20	11 (5–48)	3 (2–6), prior allo: 10 (50%)	90% (12mo)	90% (18/20)	14/18 (78%)	NR	68 (NR)	NR	50% (2/4)	14% (2/14)	NR
Turtle et al. [20]	CD19.41BB	37% Cy 57%Flu/Cy 6% Cy/Eto	30	40 (20–73)	3 (1–11), prior allo: 11 (37%)	NR	93% (27/29)	13/29 (45%)	NR	NR	NR	44% (7/16)	15% (2/13)	NR
Shadman et al. [21]°	CD19.41BB	84% Flu/Cy 16%Cy	19	39 (23–74)	3 (1–11), prior allo: 3 (16%)	58% (12) (*all had HCT*)°	NA°	19/19 (100%)°	MAC (74%) RIC (10%) NMA (16%)	72 (28–138)	58% (12)°	NA°	2/19 (11%, fatal)	4/19 NRM 21% (12mo)
Roddie et al. [22]	CD19.41BB	Flu/Cy	20	41.5 (18–62)	3 (2–6) prior allo: 13 (65%)	58% (24)	85% (17/20)	3/17 (18%)	NR	90 (90–270)	NR	36% (5/14)	33% (1/3)	NR
Roddie et al. [23]	CD19.41BB	Flu/Cy	127	47 (20–81)	2 (2–6) prior allo: 56 (44%)	61%(12) median OS 15.6mo	78% (99/127)	18/99 (18%)	NR	101 (38–421)	NR	NR	NR	NR

Legend: Abbreviations: Allo—allogeneic, AraC—cytarabin, CAR—chimeric antigen receptors, CR—complete remission, CRi—complete remission with incomplete count recovery, Cy—cyclophosphamide, DOR—duration of remission, Eto—etoposid, EFS—event-free survival, FCM-MRD—refractory minimal residual disease by flow cytometry, Flu—fludarabin, FU—followup, Haplo—haploidentical, HSCT—hematopoietic stem cell transplantation, LFS—leukemia-free survival, MAC—myeloablative conditioning, MRD—minimal residual disease, NA—not applicable, NMA—nonmyeloablative conditioning, NRM—nonelapse mortality, NR—not reported, OS—overall survival, Ped—pediatric patients, PFS—progression free survival, RFS—relapse free survival, r/r—relapse and refractory, RIC—reduced intensity conditioning, TRM—treatment related mortality, Tx—transplantation; * relapse of responders; ** includes relapse and death; *** within the two groups: (A) r/r (n = 42) 11 (3–68), (B) refractory FCM-MRD+ (n = 9) 24 (2–44); **** 6 months after HSCT; ***** patients in CR, in patients in CRi (n = 3) 134.0 days (65–175); °: retrospective analysis of HCT patients only.

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
