# Peer review of "The Changing Role of Allogeneic Stem Cell Transplantation in Adult B-ALL in the Era of CAR T Cell Therapy"

_curroncol, 2025, doi:10.3390/curroncol32030177_

Round 1

Reviewer 1 Report

Comments and Suggestions for Authors

This is a well-written review on the current evidence for allogeneic HCT in the context of CAR-T therapy for B-cell precursor ALL in adults, with a major focus on the question whether consolidating HCT should be offered after achieving CR following CAR-T. Limitations of this evidence are outlined and explained (e.g., lack of randomisation, small sample size, differences in patient cohorts, in particular regarding a previous HCT). This review is concluding that most studies revealed a PFS benefit for a consolidating HCT post CAR-T, but a significant OS benefit has not yet been shown. Thus, it is suggested to refer to established risk factors for the decision wich patients should receive HCT upon CAR-T induced remission, and which patients should not. As an established risk factor, (molecular) MRD post CAR-T, but also a novel, CAR-T related risk factor, i.e. the early loss of B cell aplasia, may guide the decision, in addition to risk factors that are already known at the time of CAR-T.

Minor:

This sentence is not understandable to me (line 57), particularly with respect to alloHCT as an induction treatment, and should be clarified: "While allo-HCT may serve as induction, consolidation, or relapse treatment, most of the 57 evidence regarding HCT following CD19-CAR T cell therapy is derived from retrospective analyses of clinical trial outcomes."      

Author Response

Comment 1: This is a well-written review on the current evidence for allogeneic HCT in the context of CAR-T therapy for B-cell precursor ALL in adults, with a major focus on the question whether consolidating HCT should be offered after achieving CR following CAR-T. Limitations of this evidence are outlined and explained (e.g., lack of randomization, small sample size, differences in patient cohorts, in particular regarding a previous HCT). This review is concluding that most studies revealed a PFS benefit for a consolidating HCT post CAR-T, but a significant OS benefit has not yet been shown. Thus, it is suggested to refer to established risk factors for the decision which patients should receive HCT upon CAR-T induced remission, and which patients should not. As an established risk factor, (molecular) MRD post CAR-T, but also a novel, CAR-T related risk factor, i.e. the early loss of B cell aplasia, may guide the decision, in addition to risk factors that are already known at the time of CAR-T.

Minor: This sentence is not understandable to me (line 57), particularly with respect to allo-HCT as an induction treatment and should be clarified: "While allo-HCT may serve as induction, consolidation, or relapse treatment, most of the 57 evidence regarding HCT following CD19-CAR T cell therapy is derived from retrospective analyses of clinical trial outcomes."     

Response 1:

  • We appreciate your time and thank you for your supportive feedback. We agree that the sentence (page 2 line 57f) was unclear. Our intention was to convey that allo-HCT is not only used as a consolidative strategy (the focus of our review) but is also utilized as a treatment option for relapse. We have revised the sentence (and paragraph) for clarity and removed "induction" since it is not typically applicable in this context (page 2 lines 58-65).

Reviewer 2 Report

Comments and Suggestions for Authors

CD19 CAR-T cell therapy has indeed ushered in a revolution in the treatment of relapsed/refractory precursor B acute lymphoblastic leukemia. Unfortunately, the majority of adults do not sustain the remission they achieve with the CAR-T therapy. Consolidation with allogenic hematopoietic stem cell transplantation (HCT) provides an option to prevent the relapse but is fraught with high TRM. The authors have put in good effort at reviewing published literature to bring out this manuscript. I find it well researched and well written. I have a few comments:

This is an excellent narrative review. However, it would add value mentioning the methodology used to select the articles reviewed. A recent publication from the City of Hope Center for Leukemia Research on their experience of 45 patients that underwent consolidative HCT observed relatively low early mortality. It would have been a good article to critique as well.

When comparing post CAR-T survival of patients who underwent consolidative HCT versus who did not, it would help to know how many in the non HCT arm were alive and relapse free at the median time to HCT for patients that underwent HCT. Landmark analyses providing such time-point comparisons provide more granular detail on the survival outcomes observed.

In the manuscript, predictive factors for post CAR-T relapse have been elegantly addressed. However, I find this addressed in the discussion and conclusion section. I believe the relevant discussion from the review of literature would be more appropriate in section 2 MAIN/Review of Literature and final discussion in the Discussion/Conclusion section.

Author Response

Comment 1: This is an excellent narrative review. However, it would add value mentioning the methodology used to select the articles reviewed.

Response 1:  Thank you for your thoughtful and encouraging feedback. We have now included a Materials and Methods section (page 2). As we were unsure whether to include such a section based on similar reviews and the author instructions from the Journal, we would like to leave it to the reviewers and editors to determine whether its inclusion is appropriate in the final version.

Comment 2: A recent publication from the City of Hope Center for Leukemia Research on their experience of 45 patients that underwent consolidative HCT observed relatively low early mortality. It would have been a good article to critique as well.

Response 2: We believe you are referring to Aldoss et al. (Transplantation and Cellular Therapy, vol. 30, 2024). We appreciate this suggestion, as this study provides valuable insights into consolidative allo-HCT following CAR T therapy. It was previously briefly mentioned in the text (reference 23) but was not included in the table. Additionally, it may have been unclear that we were referring to two different studies by the same first author (Aldoss et al. 2023 vs. Aldoss et al. 2024). To address this, we have now added the work to the table, included the statement about “relatively low early transplant-related mortality” with following details (page 5, line 217f), and have now explicitly included the publication years when referencing these studies (page 5, lines 204 and 214 and table 1). Examples of NRM rates were also updated in the abstract (page 1 line 22) and discussion (page 8 line 389).

Comment 3: When comparing post CAR-T survival of patients who underwent consolidative HCT versus who did not, it would help to know how many in the non HCT arm were alive and relapse free at the median time to HCT for patients that underwent HCT. Landmark analyses providing such time-point comparisons provide more granular detail on the survival outcomes observed.

Response 3: Thank you for your careful review. We fully agree that a landmark analysis would provide a more balanced comparison between patients who underwent HCT and those who did not by reducing the immortal time bias. While the median time to allo-HCT could be extracted from most studies (included in table 1), the original studies were typically not designed to address this specific question, and thus rarely performed landmark analyses. We reviewed all studies  listed in table 1 and confirmed that, at most, four out of the 18 studies performed (an analysis resembling) a landmark analysis. For your interest, we have attached a separate table reviewing these studies (Table 2, for review purposes only). Pan et al. do not clearly state it as such but do an analysis “at a later timepoint” of 90 days (with a median time to HCT of 84 days). The trend in their result is not significantly different from an earlier analysis (more relapses in the non-HCT group) with fewer patients (1/17 patient relapsed in HCT group and 3/5 patients relapsed in non-HCT group), which is why we did not include it in table 1. Only Frey et al. and Jiang et al. clearly use “the maximum time of HCT”. Aldoss et al. (2023) perform their analysis at day 28, although the median time to HCT is longer (79 days).

We have now explicitly acknowledged this limitation and emphasized the importance of structured landmark analyses in future studies (page 7 line 351f).

Comment 4: In the manuscript, predictive factors for post CAR-T relapse have been elegantly addressed. However, I find this addressed in the discussion and conclusion section. I believe the relevant discussion from the review of literature would be more appropriate in section 2 MAIN/Review of Literature and final discussion in the Discussion/Conclusion section.

Response 4: Thank you for pointing this out. While we do not claim it to be a full review to the level of detail as the main focus of this work, as per the reviewer’s suggestion, we have moved the paragraph on predictive factors to the Main section under a dedicated subheading (page 7, “Risk factors for relapse after CAR T therapy”), while only a concise summary is retained in the discussion.
